# FunGrasp: Functional Grasping for Diverse Dexterous Hands

**Linyi Huang[1*], Hui Zhang[2*], Zijian Wu[1], Sammy Christen[2], Jie Song[1,2,3]†**
[1]The Hong Kong University of Science and Technology (Guangzhou), [2]ETH Zürich
[3]The Hong Kong University of Science and Technology
*Equal contribution. †Corresponding author: `jsongroas@hkust-gz.edu.cn`.

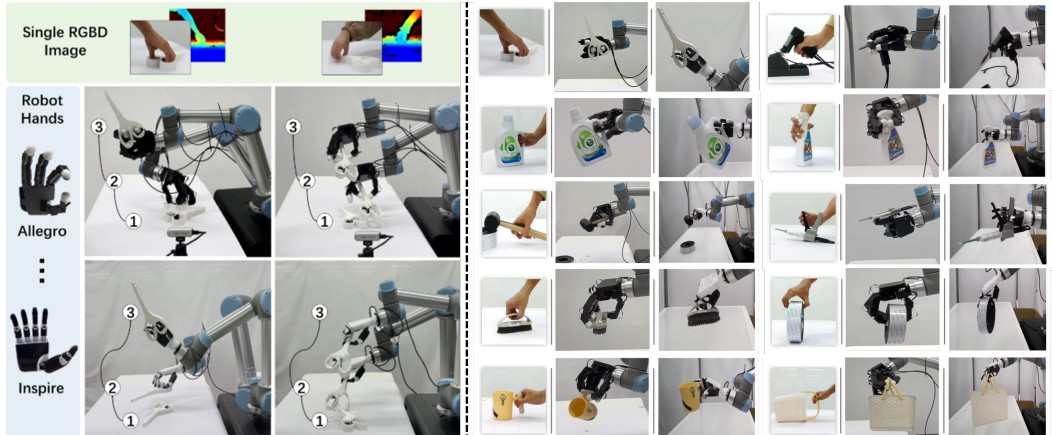

Figure 1: Our method achieves task-specific functional dexterous grasping for different robot hands with single human grasp RGBD images as input.

**Abstract:** Functional grasping is essential for humans to perform specific tasks, such as grasping scissors by the finger holes to cut materials or by the blade to safely hand them over. Enabling dexterous robot hands with functional grasping capabilities is crucial for their deployment to accomplish diverse real-world tasks. Recent research in dexterous grasping, however, often focuses on power grasps while overlooking task- and object-specific functional grasping poses. In this paper, we introduce *FunGrasp*, a system that enables functional dexterous grasping across various robot hands and performs one-shot transfer to unseen objects. Given a single RGBD image of functional human grasping, our system estimates the hand pose and transfers it to different robotic hands via a human-to-robot (H2R) grasp retargeting module. Guided by the retargeted grasping poses, a policy is trained through reinforcement learning in simulation for dynamic grasping control. To achieve robust sim-to-real transfer, we employ several techniques including privileged learning, system identification, domain randomization, and gravity compensation. In our experiments, we demonstrate that our system enables diverse functional grasping of unseen objects using single RGBD images, and can be successfully deployed across various dexterous robot hands. The significance of the components is validated through comprehensive ablation studies.

**Keywords:** Dexterous Manipulation, Grasping, Reinforcement Learning

## 1 Introduction

Humans naturally consider task-specific functions when grasping objects, such as holding a mug by its handle when drinking and by its body when washing it. Enhancing dexterous robot hands with human-like functional grasping capabilities and the ability to quickly adapt to new objects could effectively support humans in various areas, from healthcare to everyday household tasks.

Submitted to the 9th Conference on Robot Learning (CoRL 2025). Do not distribute.

Achieving such kind of human-like functional grasping capabilities brings up several challenges. First, functional grasping requires guidance from humans such as providing task-specific poses, which are difficult to transfer to dexterous robot hands due to differences in morphology, including finger numbers, knuckle sizes, and degrees of freedom (DoF). As a result, most existing works focus on reaching stable power grasps, often overlooking the integration of human guidance for achieving diverse functional poses [1, 2, 3]. Second, the robot must be able to handle various object shapes and generalize to unseen objects, which requires an efficient and general method for capturing object shape features. Nevertheless, previous methods often train category-level policies with limited generalization ability to novel objects [2, 3]. Finally, robot finger motors have limited power and precision due to size constraints, which makes them susceptible to disturbances and complicates sim-to-real transfer due to inaccurate joint dynamic models. As a result, current dexterous grasping approaches that accommodate diverse poses and objects remain largely confined to simulation, lacking validation for their sim-to-real capabilities[4, 5, 6, 7]. Overall, a functional dexterous robot grasping system capable of grasping diverse unseen objects in a human-like manner in real-world settings is still missing.

In this paper, we present *FunGrasp*, a system for functional dexterous robot grasping that utilizes task-specific human grasping poses as priors. By leveraging single RGBD images of human grasps, our system achieves one-shot generalization to unseen objects. Furthermore, it can be deployed on various robotic hand platforms. Our system consists of three stages introduced in Section 2: static functional grasp retargeting, dynamic dexterous grasping, and sim-to-real transfer.

In our experiments, we first demonstrate that our system can successfully achieve task-specific functional dexterous robot grasping of unseen objects in the real world, given single RGBD images of human grasps. We further provide rich qualitative results to showcase the diversity of the generated task-specific functional robot grasps. Next, we evaluate the generalization ability of our system across different dexterous robot hands in both simulation and real-world settings. Finally, we conduct an ablation study on the components of our system to demonstrate their effectiveness.

In summary, our contributions are: 1) *FunGrasp*, a system that achieves functional dexterous robot grasping in the real world and performs one-shot generalization to unseen objects from a single RGBD image of a human grasp. 2) A retargeting module that effectively transfers task-specific functional grasp poses from humans to diverse dexterous robot hand models while preserving both human-like postures and precise contact points. 3) A system identification module that provides accurate joint dynamic models for dexterous robot hands, facilitating robust sim-to-real transfer. 4) Experiments demonstrating that our system can effectively generalize to various dexterous robot hands in both simulation and real-world settings.

## 2 Methods

In this paper, we address the challenge of functional dexterous robot grasping. To effectively draw guidance from human grasping behavior, we utilize a single RGB-D image to extract a static reference of a functional human hand grasp $\mathbf{G}_h = (\overline{\mathbf{q}}_h, \overline{\mathbf{T}}_h, \overline{\mathbf{T}}_o, \overline{\mathbf{c}})$, where $\overline{\mathbf{T}}_h$ and $\overline{\mathbf{T}}_o$ represent the 6D global hand and object poses, respectively, while $\overline{\mathbf{q}}_h$ indicates the target finger joint angles, and $\overline{\mathbf{c}}$ specifies the target binary contact states of each finger link with the object which are derived by distances. We assume $\mathbf{G}_h$ can be obtained from existing datasets or extracted utilizing off-the-shelf pose estimators. With the given $\mathbf{G}_h$, the goal is to accordingly control the robot hand with an arm to grasp the object in a human-like functional manner. This is accomplished through the observation of the wrist 6D pose $\mathbf{T}_r$ and velocity $\dot{\mathbf{T}}_r$, object 6D pose $\mathbf{T}_o$ and velocity $\dot{\mathbf{T}}_o$, and robot hand finger joint angles $\mathbf{q}_r$.

Fig. 2 outlines our system, which comprises three modules: (A) H2R Grasp Retargeting, (B) Dynamic Grasp Control, and (C) Sim-to-Real Transfer. Given the human hand grasp reference $\mathbf{G}_h$, we first retarget it to a static robot hand grasp reference $\mathbf{G}_r$, ensuring that the human-like posture and precise contact positions are preserved. Next, we train a policy using reinforcement learning (RL) in a simulation environment to enable the robot hand to perform dynamic grasping in accordance with $\mathbf{G}_r$. Finally, we transfer the policy developed in simulation to real robot hands.

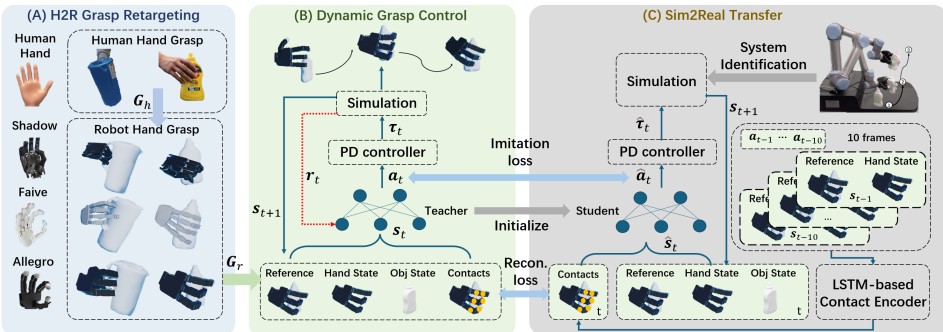

Figure 2: System Overview.

**1) Static Functional Grasp Retargeting:** We utilize off-the-shelf hand-object pose estimation models [8, 9] to obtain functional human grasp poses from single RGBD images with known object meshes. To effectively transfer these functional grasping poses to dexterous robot hands, we propose a retargeting module without requirements for specific hand morphologies. We initialize the robot hand poses with the human grasping poses by aligning the corresponding link directions in the object frame. Then we optimize the poses to preserve precise contact positions and human-like postures for functional grasping. We also consider the undesired collision and penetrations, joint constraints, and force closure grasps during optimization.

**2) Dynamic Dexterous Grasping:** We adopt a reinforcement learning (RL) framework to achieve dynamic functional grasping for diverse dexterous robot hands from static grasping poses. To handle diverse object shapes with a single policy and generalize to unseen object geometry, we utilize the implicit object shape feature similar to [4], described by the grasp pose reference and local contact information, as the target joint positions and contacts indicate the object's local shape around the contact points. The robot hand further leverages proprioception and real-time hand-object contact states, which can be either privileged information or reconstructed from proprioception, to refine the implicit perception of object shapes.

**3) Sim-to-Real Transfer:** We utilize several techniques to enable effective sim-to-real transfer. Specifically, we utilize privileged learning that distills the policy trained with privileged contact information into a policy that relies on the information available in the real world. We employ system identification to model accurate actuator dynamics by optimizing the joint stiffness and damping factors. We first train a grasping policy in simulation with rough initial parameter values. The policy is then deployed on the hardware in an open-loop manner to gather diverse action-state trajectories. Then we apply the recorded action trajectories in the simulation and optimize the parameters by minimizing the discrepancies between the states in the simulation and the states recorded from the hardware. Finally, we fine-tune the pre-trained policy with the optimized parameters.

## 3 Experimental Results

Our default setup uses a UR5 [10] robotic arm equipped with an Allegro Hand [11] as shown in Fig. 3. Additionally, we utilize the Inspire Hand [12] to verify our system's generalization ability across morphologies. We deploy a static RealSense D435i camera to perform object tracking.

We follow the definitions of metrics provided in [4, 5]. **1) Success Rate (Suc. R.)**: A grasp is considered successful if the object can be lifted higher than 0.1 m and does not fall for 3 seconds. **2) Simulated Distance (SimD.)** (simulation only): Similar to [4], we report the mean displacement of the object in mm per second to evaluate the stability of the grasp. **3) Contact Ratio (Con. R.)** (simulation only): To evaluate the precision of the actual grasps, we measure the ratio between the achieved contacts in the simulation and the target contacts defined via the grasp reference $\mathbf{G}_r$.

### 3.1 Generalization Across Diverse Robot Hands

To demonstrate our generalization ability across various dexterous robot hands, we conduct a quantitative evaluation with three robotic hands [11, 13, 14] in simulation and further perform qualitative assessments on two real robot hands [11, 12] in the real world. We train a policy for each hand

| Model | Suc. R. ↑ | SimD. [mm/s] ↓ | Con. R. ↑ |
|---|---|---|---|
| Shadow Hand | 75% | 1.6 | 0.75 |
| Faive Hand | 81% | 1.9 | 0.80 |
| Allegro Hand | 85% | 1.6 | 0.79 |

Table 1: Generalization to different robot hands (Sim).

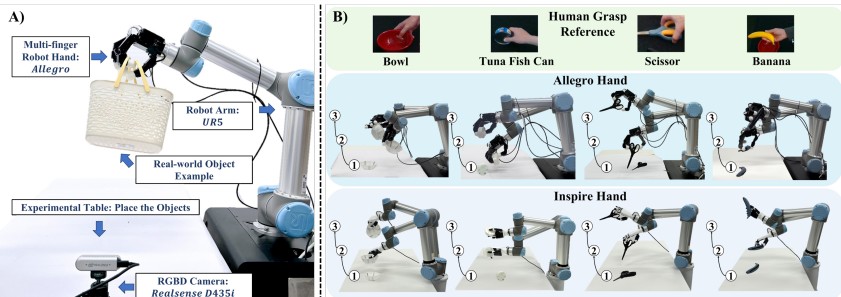

Figure 3: **A)** Hardware setup. **B)** The grasping motions of different hands with the same references.

model with the training set of 75% DexYCB grasp references and evaluate with the remaining 25% references. We 3D-print the YCB objects [15] for real-world experiments.

We first quantitatively evaluate the performance in simulation with Shadow [13], Faive [14], and Allegro [11], with the results shown in Tab. 1. Although the hands exhibit significant variations in size and morphology (e.g., DoF and finger numbers), our system achieves success rates of 75%+ consistently across all the hand models. Specifically, Allegro Hand shows the best performance due to its larger size that facilitates easier wrapping and grasping of the objects, while Shadow Hand exhibits a lower success rate, attributed to its smaller size and the limited range of its finger swing joints, making it challenging to grasp larger objects. We also qualitatively show the results on real Allegro and Inspire Hands in Fig. 3. Our system achieves diverse human-like functional grasps for the two hands. Notably, both hands can precisely follow the same human grasp references and successfully grasp thin and small objects lying on the table. This is challenging due to the potential collisions between the hands and the table, indicating the effectiveness of our system in leveraging and preserving the precise postures and contacts from human grasp references.

### 3.2 One-shot Functional Grasping of Unseen Objects

We evaluate the one-shot generalization capability of our system for task-specific functional grasping of unseen objects in the real world, using single RGBD images of human grasps. We select 20 common daily objects for evaluation as shown in Fig. 4, which were not seen by the policy during training. For each reference, we place the object on the table with two different poses and perform grasps, resulting in six grasps per object.

Our method achieves comparable high success rates for both hands in the real world, with 73% for Inspire Hand and 74% for Allegro Hand. It can successfully grasps unseen objects with diverse shapes, sizes, and masses, ranging from a long, heavy hammer to a large, light basket. Notably, it can effectively grasp a deformable loopy doll that has completely different physical features from the training objects. The results verify the generalization ability of our system. Fig. 1 and Fig. 5 shows various human-like functional grasps of our system with provided human grasp RGBD images.

## 4 Limitations and Conclusion

In this work, we present *FunGrasp*, a system capable of performing one-shot functional dexterous robot grasping of unseen objects from single human grasp RGBD images. We have developed a comprehensive system that transitions from human grasp images to dexterous robot dynamic functional grasping, demonstrating generalization across diverse task-specific grasping poses, various object shapes, and different dexterous robot hands. A current limitation of our system is its reliance on known object meshes for off-the-shelf pose estimation models, which are used to extract human grasping poses from RGBD images and obtain object state observations from the camera. An integrated model with image inputs could enhance further generalization capabilities. Finally, our system cannot handle robot hands with rather different morphologies such as two-finger grippers.

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

# A Extended Related Work

For a better comparison of existing dexterous robot grasping works and ours, we list the differences in Tab. 2.

Table 2: Comparison with existing dexterous robot grasping works. Our method achieves functional grasping with diverse task-specific poses, and can generalize to diverse real robot hands and unseen objects.

| Method | Hardware Deployment | Functional Grasping | Cross-category Generalization | Diverse Robot Hands | Diverse Poses |
|---|---|---|---|---|---|
| D-Grasp [4] | ✗ | ✓ | ✓ | ✗ | ✓ |
| UniDexGrasp [6] | ✗ | ✗ | ✓ | ✗ | ✓ |
| UniDexGrasp++ [7] | ✗ | ✗ | ✓ | ✗ | ✗ |
| GraspXL [5] | ✗ | ✓ | ✓ | ✓ | ✓ |
| DexPoint [2] | ✓ | ✓ | ✗ | ✗ | ✗ |
| DexTransfer [16] | ✓ | ✓ | ✗ | ✗ | ✓ |
| Agarwal et al.[3] | ✓ | ✓ | ✗ | ✗ | ✗ |
| FunGrasp (Ours) | ✓ | ✓ | ✓ | ✓ | ✓ |

## A.1 Dynamic Dexterous Grasping

Dexterous manipulation is a long-standing research topic in robotics [2, 3, 4, 5, 17, 18, 19, 20, 21, 22, 23, 24, 25, 26, 27, 28, 29]. Among all the manipulation tasks, dexterous grasping is one of the most fundamental skills [4, 5, 6, 7, 27, 28, 29, 30, 31, 32, 33, 34]. Some studies have attempted to use reinforcement learning to train dynamic dexterous grasping policies [4, 5, 6, 7, 35] and have shown promising results in simulation. Specifically, we follow [4] to design the dynamic dexterous grasping module, as [4] is verified to be effective to train dynamic grasping policies from static references. However, compared to our system, their method focuses on generating human hand grasping motions without considering different robot hand morphologies and hardware deployment. Additionally, their policy is trained with access to ground-truth states, which are not accessible on a real robot. Some studies attempt to predict the static grasping poses and deploy them on real robots [27, 28], but these approaches lack the dynamic adaptation ability to disturbances due to open-loop static-pose execution. Some other works [2, 3] achieve dynamic dexterous grasping on real robots with RL, but focus on category-level policies and therefore only generalize within the same categories. In contrast, our system can deal with diverse objects with one single policy and generalize to unseen categories through the use of an image prior. Besides, most of the above-mentioned methods focus on power grasps without considering the functional aspects of the objects, while our system can achieve diverse human-like task-specific functional grasping poses precisely.

Recently, some works have utilized teleoperation to collect data directly on real robots for imitation learning [36, 37, 38], which fundamentally solves the problem caused by the sim-to-real gap. Furthermore, through the collected motion trajectories, these methods can deal with more manipulation tasks beyond grasping. However, the expensive teleoperation process leads to limited available data, and the data collected with one hand model cannot be used for another hand, which further increases the cost of data collection and limits their generalization ability. In contrast to these works, our system has a specific focus on dexterous grasping. We utilize RL to achieve robust grasping on real robots without real robot manipulation data, and can be deployed on various dexterous robot platforms with no hand morphology assumption.

## A.2 Human-to-Robot Grasp Retargeting

Enabling robot hands with human-like manipulation capabilities requires prior knowledge about the way in which humans interact with environments and manipulate objects, such as how to grasp

objects in a task-specific functional manner. Recent works attempt to extract such kind of prior from human manipulation data by retargeting the human grasping poses to dexterous robot poses [16, 39, 40, 41, 42, 43]. Some works focus on mimicking the hand postures by mapping the corresponding joint angles from the human hand to the robot hand [41, 44]. These works can efficiently preserve human-like grasp postures but usually lead to inaccurate contacts, which limits their precision in functional grasping. Instead of joint-to-joint retargeting, some other works focus on the fingertips which are usually more important for achieving robust grasping [45, 46, 47]. These methods simplify the retargeting process as they only consider fingertip positions, which makes them easier to deploy on different robot hands. However, the simplified retargeting leads to inaccurate postures without human likeness which sometimes even breaks the joint limits, making downstream joint control difficult. In contrast, with consideration of both human-like postures and precise contact points, our retargeting module can effectively maps human grasping poses to different dexterous robot hands to grasp objects in a task-specific functional manner.

# B Extended Methods

## B.1 H2R Grasp Retargeting

Robot hands exhibit a variety of structures that differ from human hands, including variations in DoF, finger numbers, and knuckle sizes. Consequently, the human hand grasp reference $\mathbf{G}_h$ should be retargeted to a robot hand reference $\mathbf{G}_r$ before it can effectively guide robotic grasping. Motivated by this, we first initialize $\mathbf{G}_r$ with the same fingertip positions and finger link directions as $\mathbf{G}_h$ in the object frame. For robot hands with fewer fingers (e.g., Allegro Hand [11]) or finger joints (e.g., Inspire Hand [12]), we simply remove the pinky finger or joints close to the fingertips.

After initialization, we optimize the retargeted pose by considering the hand-object interaction with several losses. We utilize the penetration energy loss $L_{\text{pen}}$ from [48] and force closure loss $L_{\text{fc}}$ from [49] to avoid hand-object penetration and encourage stable grasping. Additionally, we introduce the contact position loss $L_{\text{pos}}$ to incentivize the robot hand to remain in contact with the object at the right position, with the formulation $L_{\text{pos}} = \sum_{\bar{\mathbf{c}}_j=1} \left\| \mathbf{p}_j^h - \mathbf{p}_j^r \right\|^2$, where $\mathbf{p}_j^h$ and $\mathbf{p}_j^r$ are the positions of the $j_{th}$ human hand joint and its corresponding robot hand joint, and $\bar{\mathbf{c}}_j = 1$ indicates that the $j_{th}$ human hand joint is in contact with the object. To regularize the joint angles, we apply the limit loss $L_{\text{joints}} = \sum_{i=1}^{M} \left( \max(0, \theta_i - \theta_i^{\text{upper}}) + \max(0, \theta_i^{\text{lower}} - \theta_i) \right)$, where $M$ is the number of robot hand joints, $\theta_i^{\text{lower}}$ and $\theta_i^{\text{upper}}$ are the lower and upper limits of the $i_{th}$ joint. Finally, we introduce a collision loss $L_{\text{col}} = \sum_{i=1}^{M} (\sum_{j=1|(i \neq j)}^{M} \max(\tau - d(i,j), 0) + \max(h_{i-\text{table}}, 0))$ to punish the collision between the robot hand with itself and the table, where $M$ is the number of robot hand joints, $d(i,j)$ is the distance between the $i_{th}$ and $j_{th}$ joints, $\tau$ is a threshold, and $h_{i-table}$ is the signed distance from $i_{th}$ joint to the table surface.

## B.2 Dynamic Grasp Control

Following [4], we formulate dynamic grasp control guided by the retargeted pose reference $\mathbf{G}_r$ as a reinforcement learning problem. We remove the wrist guidance from [4] in both simulation and real-world deployment, as the additional torques applied to the hand wrist can lead to excessively rapid hand movements, which may compromise safety.

### B.2.1 Network Structure

In simulation, the state space $s = (\mathbf{q}_r, \mathbf{T}_r, \dot{\mathbf{T}}_r, \mathbf{T}_o, \dot{\mathbf{T}}_o, \mathbf{c}, \mathbf{f}, \mathbf{G}_r)$ includes the robot's joint angles $\mathbf{q}_r$, the hands's 6D global wrist pose $\mathbf{T}_r$ and its velocity $\dot{\mathbf{T}}_r$, the object's 6D pose $\mathbf{T}_o$ and its velocity $\dot{\mathbf{T}}_o$, per finger part binary contact states $\mathbf{c}$ and contact forces $\mathbf{f}$, as well as the reference $\mathbf{G}_r$. A feature extraction layer $\phi$ is applied such that $\phi(\mathbf{s}) = (\mathbf{q}_r, \tilde{\mathbf{T}}_r, \dot{\tilde{\mathbf{T}}}_r, \tilde{\mathbf{T}}_o, \dot{\tilde{\mathbf{T}}}_o, \tilde{\mathbf{p}}_o, \tilde{\mathbf{p}}_r^z, \mathbf{f}, \tilde{\mathbf{g}}_p, \tilde{\mathbf{g}}_r, \mathbf{g}_c)$. $\tilde{*}$ denotes variables expressed in the initial wrist frame, which helps the policy concentrate on hand-object interactions without being affected by global poses as verified in [4]. $\tilde{\mathbf{p}}_o$ and $\tilde{\mathbf{p}}_r^z$ represent the object displacement and the wrist-table distance, respectively. $\tilde{\mathbf{g}}_p$ indicates the distance between the current

and the target 3D position of each joint, while $\tilde{\mathbf{g}}_r$ represents the difference between the current and target wrist rotations. The term $\mathbf{g}_c = [\bar{\mathbf{c}}|\bar{\mathbf{c}} - \mathbf{c}]$ contains the binary target contacts and the difference between the target and the current contacts. With the extracted features as inputs, the policy outputs the action $\mathbf{a}$, which is defined as the predicted finger joint angles and wrist 6D poses for the next frame. The predicted wrist 6D poses are used to calculate the arm joint angles through inverse kinematics, and the arm and finger joint angles are further fed into PD controllers to compute the joint torques.

### B.2.2 Reward Function

To incentivize the policy to learn the desired behavior, we define the reward function as $r = \omega_p r_p + \omega_c r_c + \omega_s r_s + \omega_q r_q$. It comprises the joint position reward $r_p$, contact reward $r_c$, safety reward $r_s$, and pose reward $r_q$. Inspired by [4], we adopt the same formulation for $r_p$ and modify $r_c$ with a dynamic weight $\omega_c = (\sum_{\bar{\mathbf{c}}_j=1} ||\mathbf{p}_j^r||^2)/(\sum_{\bar{\mathbf{c}}_j=1} ||\bar{\mathbf{p}}_j^r||^2)$ to facilitate accurate contact positions, where $\mathbf{p}_j^r$ and $\bar{\mathbf{p}}_j^r$ represent the current and target contact positions. The safety reward $r_s = \sum_{i=1}^L |f_{\text{colli}}^i|$ penalizes the undesired contact forces of the hand with the table and itself, where $L$ is the number of links and $f_{\text{colli}}^i$ represents the undesired collision force of the $i_{th}$ link. The pose reward $r_q$ encourages the robot's hands to maintain human-like postures, defined as $r_q = \frac{1}{F \cdot K} \sum_{i=1}^F \sum_{j=1}^K (\frac{\mathbf{v}_{ij}\bar{\mathbf{v}}_{ij}}{|\mathbf{v}_{ij}||\bar{\mathbf{v}}_{ij}|} - 1)$ where $F$ is the number of fingers, $K$ is the number of links per finger, $\mathbf{v}_{ij}$ and $\bar{\mathbf{v}}_{ij}$ are the current and target directions of the $j_{th}$ link on the $i_{th}$ finger in the object frame.

### B.3 Sim-to-Real Transfer

Our work focuses on real robot hand grasping, making sim-to-real transfer a crucial aspect. To achieve this, we adopt a privileged learning framework to learn an effective policy without privileged information, utilize system identification methods to accurately model the robot's joint dynamics, apply domain randomization to enhance model robustness, and incorporate gravity compensation to address the effects of hand gravity.

### B.3.1 Privileged Learning

Tactile observation is crucial for learning robust functional grasping; however, such information is not available on real robots. Additionally, training the grasping policy while reconstructing tactile information in an end-to-end manner poses significant challenges, as proper contacts require effective grasping while effective grasping requires contact observation. To address this, we employ a privileged learning framework. First, we train a teacher policy based on an MLP using ground-truth contact information obtained from simulation through reinforcement learning. Subsequently, we distill a student policy that utilizes only the information accessible in the real world. Specifically, we train the student policy incorporating an additional LSTM-based encoder to reconstruct the contacts from proprioceptive data while simultaneously imitating the teacher's actions. The MLP of the student policy is initialized with the weights of the teacher. Specifically, the encoder takes state-action pairs of the past 10 frames as inputs, which include the finger joint angles $\mathbf{q}_r$, the difference between the current and target wrist 6D poses $\mathbf{T}_r$ and $\bar{\mathbf{T}}_r$, the target binary contact states $\bar{\mathbf{c}}$, and the actions $\mathbf{a}$. It then predicts the contacts ($\hat{\mathbf{c}}_t$ and $\hat{\mathbf{f}}_t$), which are subsequently input into the MLP alongside the other observations to predict the actions $\hat{\mathbf{a}}_t$. Intuitively, the LSTM encoder can infer the torques applied to the joints through the actions and the state changes through the joint angle trajectories. The misalignment between the joint torques and state changes can further indicate external forces caused by contacts, which can be used to infer the contact states and forces of the finger links. To train the student policy, the contact reconstruction loss is defined as $L_{\text{re}} = ||\hat{\mathbf{c}}_t - \mathbf{c}_t||^2 + ||\hat{\mathbf{f}}_t - \mathbf{f}_t||^2$, while the action imitation loss is given by $L_{\text{act}} = ||\hat{\mathbf{a}}_t - \mathbf{a}_t||^2$.

### B.3.2 System Identification

To narrow the sim-to-real gap, we model the finger joint dynamics by identifying the actuator parameters of the real robot hand, including joint stiffness and joint damping. Specifically, we first

pre-train a policy in simulation using rough initial values for these parameters. We then deploy this policy on the real robot in an open-loop manner to collect command-state trajectories. Next, we align the simulated and real hand state trajectories based on the collected action trajectories by optimizing the parameters in simulation. Finally, we fine-tune our pre-trained policy using the optimized parameters. We adopt the CMA-ES[50] method to optimize the parameters in simulation using the MSE loss, defined as $L_{\text{Sim-Real}} = \sum_{t=1}^{N} \|\mathbf{q}_t^s - \mathbf{q}_t^r\|^2$, where $\mathbf{q}_t^s$ and $\mathbf{q}_t^r$ represent the robot joint angles in simulation and in the real world under the same action at time step $t$, respectively, and $N$ denotes the trajectory length.

### B.3.3   Domain Randomization

To achieve robust sim-to-real transfer, we employ domain randomization during the training of our policies, similar to other works in the field [24, 51]. Specifically, we randomize several parameters, including the damping of each joint, the gains for the PD controller, the friction coefficients, the mass of the objects, the height of the table, and the hand state observations.

### B.3.4   Gravity Compensation

To account for the effects of hand gravity on each finger joint, we calculate the physical attributes of each robot hand link, including mass distribution and center of mass locations, using the open-source Kinematics and Dynamics Library [52]. We then compute the gravity-induced torques on each finger joint in real-time based on current hand states, and compensate these torques with feedforward terms to the actuators. This module ensures precise maintenance of intended trajectories and postures by effectively compensating for gravitational forces in real time.

## C   Extended Experiments

### C.1   Experiment Details

#### C.1.1   Implementation Details

We use RaiSim [53] as the simulation engine and PPO [54] for RL training. We train the policy using a single NVIDIA RTX 3090 GPU and 128 CPU cores, which takes approximately two days. We set the initial finger joint angles with $0.5 \cdot \overline{\mathbf{q}}_r$, where $\overline{\mathbf{q}}_r$ is the target finger joint angles indicated by $\mathbf{G}_r$. The arm joint angles are initialized by the inverse kinematics (IK) solver [55] to make the wrist 30 cm away from the object center along the direction from the object center to the target wrist position. For hardware deployment, the arm and finger joints move to the initial angles under position control, and then a PD controller is deployed with a frequency of 10 Hz for grasping according to the policy and IK solver outputs. We limit the end-effector velocity to be smaller than 0.25 m/s and acceleration smaller than 0.3 m/s$^2$ for safety.

#### C.1.2   Data

We utilize the right-handed sequences from the DexYCB dataset [56] for training. Specifically, we get the human hand grasp reference $\mathbf{G}_h$ from the hand-object state of the frame when the object displacement exceeds a predefined threshold. We use 75% of the data for training and 25% of the data for testing. We utilize FoundationPose [8] to estimate the object 6D pose $\mathbf{T}_o$ and velocity $\dot{\mathbf{T}}_o$ of the objects with known object meshes. Additionally, we apply a low-pass filter to reduce the jitter in the estimated poses.

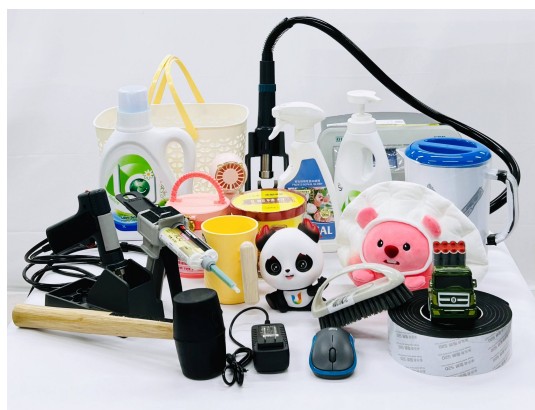

Figure 4: Objects used for one-shot generalization evaluation.

| Hand Model | Inspire Hand | Allegro Hand |
|:---:|:---:|:---:|
| Success Rate | 73% | 74% |

Table 3: One-shot generalization to unseen objects (Real).

## C.2 One-shot Functional Grasping of Unseen Objects

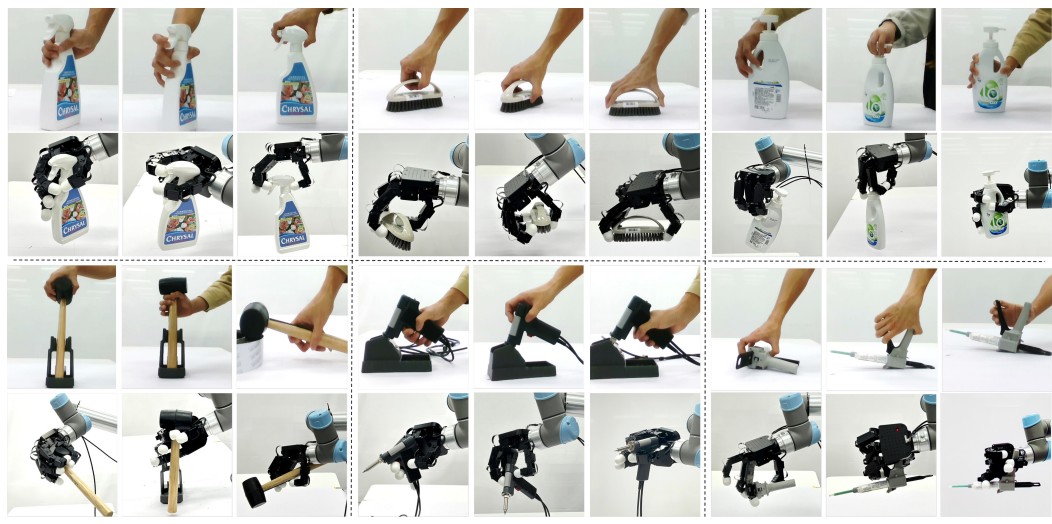

Figure 5: Diverse functional grasps from single RGBD images.

We select 20 common daily objects for evaluation as shown in Fig. 4, which were not seen by the policy during training. We use a commercial 3D scanner to obtain the meshes. For each object, we capture three RGBD images of human grasps with different poses and further utilize FoundationPose [8] to estimate the object pose, along with Metro [9] to estimate the hand poses. This process results in three human grasp references, denoted as $\mathbf{G}_h$. For each reference, we place the object on the table with two different poses and perform grasps, resulting in six grasps per object. The results are presented in Tab. 3.

## C.3 Robustness Evaluation

To verify the robustness of our system, we evaluate our method with extra noises added to the human reference poses $\mathbf{G}_h$, using the unseen objects shown in Section C.2. Specifically, we introduced two

| Setting | Wrist Bias | Joint Bias | PD | PD (Finger-closing) | Ours |
|---|---|---|---|---|---|
| Success Rate | 66% | 69% | 40% | 53% | 74% |

Table 4: Robustness Evaluation (Real).

| Model | Suc. R. ↑ | SimD. [mm/s] ↓ | Con. R. ↑ |
|---|---|---|---|
| Angle Reset + * | 62% | 1.6 | 0.65 |
| DexGraspNet + * | 65% | 1.6 | 0.68 |
| DexGraspNet (Tab. Col.) + * | 74% | 1.9 | 0.70 |
| Ours | **85%** | **1.6** | **0.79** |

Table 5: H2R Grasp Retargeting module ablation (Sim).

types of random disturbances: wrist positional bias within 5 cm (Wrist Bias), and joint angle bias within 0.09 rad applied on 4 randomly selected finger joints (Joint Bias). The results are shown in Tab. 4. Our method gets success rates degraded with 8% under wrist positional bias and 5% under finger joint angular bias. The performance drop is acceptable, considering 5 cm and 0.09 rad are around 10-20% of the object dimensions and finger joint movements. It is important to note that the reference poses $\mathbf{G}_h$ are already noisy and biased without extra disturbances, due to the imperfect reconstruction from single RGBD images. We also qualitatively demonstrate the robustness of our method against external forces and real-time adaptation behaviors under dynamic disturbances in the supplementary video.

To further compare the advantage in robustness of our RL-based method, we compare the success rates of our method with non-RL-based methods, one with a PD controller to execute the reference poses $\mathbf{G}_r$ (PD) and the other one further closing the finger joints (except the swing joints) by 0.15 rads for more tight grasps (PD (Finger-closing)). The results are shown in Tab. 4. Our method shows significantly higher success rates, indicating our better robustness to the noisy reference poses $\mathbf{G}_r$. More importantly, the results verify that a simple execution of $\mathbf{G}_r$ and strategy of finger-closing are not enough for robust functional grasping, which shows the contribution of our RL-based grasping policy.

## C.4 Ablation

We conduct a comprehensive ablation study of the various components of our system to demonstrate their effectiveness in both simulation and real-world scenarios. As in the previous section, we utilize YCB objects and the 25% DexYCB grasp references unseen during training for testing.

### C.4.1 H2R Grasp Retargeting

We first verify the effectiveness of our H2R Grasp Retargeting module in simulation by replacing it with three baselines and retraining our RL policy: i) simply setting the robot hand finger angles to match those of the human hand (Angle Reset + *). ii) utilizing DexGraspNet [48] to generate $\mathbf{G}_r$. Since DexGraspNet requires hand contact points and initial wrist poses to generate grasping poses, we extract this information from $\mathbf{G}_h$ for a fair comparison (DexGraspNet + *). iii) an extension of DexGraspNet that incorporates our table collision loss (DexGraspNet (Tab. Col.) + *). As shown in Tab. 5, our system outperforms all baselines. The Angle Reset method struggles with thin or small objects without considering contacts. Although DexGraspNet creates contact-rich grasps, its lack of table collision awareness leads to failures for thin objects. Incorporating the table collision loss improves its **Suc. R.** from 65% to 74%, especially for thin objects, but also increases **SimD.** due to reduced force-closure stability. This instability arises because the optimization of DexGraspNet does not consistently yield effective and stable poses, necessitating an additional filtering process. In contrast, our system preserves precise contacts and postures from the human grasp references, enabling more robust grasping.

| Model | Suc. R. ↑ | SimD. [mm/s] ↓ | Con. R. ↑ |
|---|---|---|---|
| w/o Priv. Info. | 61% | 1.6 | 0.56 |
| w/o Priv. Learn. | 40% | 1.7 | 0.24 |
| Teacher Policy | 85% | 1.6 | 0.79 |
| Ours | 85% | 1.6 | 0.79 |

Table 6: Privileged information & encoder ablation (Sim).

| Setting | w/o Sys. Id. | w/o Grav. Comp. | Ours |
|---|---|---|---|
| Success Rate | 38% | 59% | 75% |

Table 7: System identification & gravity compensation ablation (Real).

### C.4.2 Privileged Learning

To show the effectiveness of the privileged information (**c** and **f**) and our privileged learning framework, we compare our system in simulation against i) the policy trained without **c** and **f** (*w/o Priv. Info.*) ii) the student policy with the LSTM encoder trained from scratch without the privileged learning framework, driven by RL rewards and the contact reconstruction loss (*w/o Priv. Learn.*) iii) the teacher policy with access to the ground-truth (GT) **c** and **f** from simulation (*Teacher Policy*). The results are shown in Tab. 6. Our original policy shows a clear improvement compared with the policy trained without **c** and **f**, which shows that the contact information is key for stable and robust grasping, and indicates the necessity of our LSTM privileged information encoder. The student policy trained from scratch reaches the worst performance, which shows the importance of the privileged learning framework instead of a single-stage training process. Notably, our policy shows highly comparable performance with the teacher policy utilizing the GT privileged information, which verifies the effectiveness of our LSTM-based encoder.

### C.4.3 System Identification & Gravity Compensation

We ablate the system identification (w/o Sys. Id.) and gravity compensation (w/o Grav. Comp.) techniques (See Section B.3) on real robots to assess their effectiveness for robust sim-to-real transfer. We randomly select 6 grasp references for each YCB object from the 25% DexYCB test set to evaluate all variants. The success rates are shown in Tab. 7. Notably, the variant without system identification exhibits the poorest performance across nearly all objects, highlighting the importance of a precise joint dynamics model. The gravity compensation leads to further improvements in success rate, indicating its effectiveness. Overall, our original configuration achieves the best performance, validating the effectiveness of both modules for sim-to-real transfer.

