# OpenReview forum: "FunGrasp: Functional Grasping for Diverse Dexterous Hands"
_robot-learning.org/CoRL/2025/Workshop/Dexterous_Manipulation — CoRL 2025 Workshop Dexterous Manipulation Spotlight_

### Official Review · Reviewer_McC4 · 2025-09-04
**Reviewer Report**

**Rating:** 7
**Confidence:** 3

**Review:**

This paper presents FunGrasp, a practical pipeline that converts a single human functional-grasp RGB-D example into a task-oriented grasp executed by diverse multi-finger robotic hands. The system combines: (1) a human→robot static retargeting optimization that preserves human contact intent while respecting robot kinematics and collisions; (2) an RL-based dynamic controller (PPO) that refines and executes the grasp; and (3) a sim-to-real stack including privileged teacher→student learning, system identification, domain randomization, and gravity compensation. The evaluation includes cross-hand simulation experiments and one-shot real-robot tests on previously unseen objects, together with ablations demonstrating the contribution of key components.

Strengths
1.Comprehensive, end-to-end system: addresses perception → retargeting → control → sim-to-real deployment in a single pipeline.
2.Focus on functional (task-specific) grasps rather than only power/stable grasps; preserves human contact intent.
3.Demonstrated cross-hand generalization: evaluated on multiple hand models and transferred to real hardware with one-shot unseen objects.
4.Practical sim-to-real recipe: combines privileged learning, system identification and domain randomization effectively; ablations show these components matter.

Weaknesses:
1.Dependence on object meshes and pose estimates. The pipeline assumes an available object mesh and reasonably accurate human/ object pose/contact estimation from a single RGB-D frame. This limits applicability to truly unknown objects or scenarios with severe perception failures.
2.Failure modes and limitations. The manuscript would benefit from a clearer taxonomy of failure cases (why grasps fail: contact mismatch, slipping, collisions, perception error, insufficient force closure, etc.) with representative visualizations.

---

### Official Review · Reviewer_xcCz · 2025-09-10
**Review of FunGrasp: Functional Grasping for Diverse Dexterous Hands**

**Rating:** 9
**Confidence:** 4

**Review:**

This paper presents FunGrasp, a system that lets robot hands perform human-like, task-specific grasps using just a single RGBD image of a human grasp. The key idea is to estimate the human hand pose, retarget it to different robot hands (even with very different shapes), and then use reinforcement learning to control the robot to grasp objects functionally and reliably. The system also includes techniques to transfer what’s learned in simulation to real robots.

Cool idea to leverage human grasps directly for robots, making the grasping more functional, not just stable.
The retargeting module works well across different robot hands, showing nice generalization.
Strong experiments with multiple hands and unseen objects, including real-world tests with good success rates (~75%).

It depends on having known object models for pose estimation, which limits it to familiar objects.  The perception and grasp generation are separate steps; an end-to-end approach might improve robustness.

---

### Decision · Program_Chairs · 2025-09-18

Accept (Spotlight)